# CausalDyna: Improving Generalization of Dyna-style Reinforcement Learning via Counterfactual-Based Data Augmentation

## Abstract

Deep reinforcement learning agents trained in real-world environments with a limited diversity of object properties to learn manipulation tasks tend to suffer overfitting and fail to generalize to unseen testing environments. To improve the agents' ability to generalize to object properties rarely seen or unseen, we propose a data-efficient reinforcement learning algorithm, CausalDyna, that exploits structural causal models (SCMs) to model the state dynamics. The learned SCM enables us to counterfactually reason what would have happened had the object had a different property value. This can help remedy limitations of real-world environments or avoid risky exploration of robots (e.g., heavy objects may damage the robot). We evaluate our algorithm in the CausalWorld robotic-manipulation environment. When augmented with counterfactual data, our CausalDyna outperforms state-of-the-art model-based algorithm, MBPO and model-free algorithm, SAC in both sample efficiency by up to 17% and generalization by up to 30%. Code will be made publicly available.

## 1 Introduction

Classical model-free reinforcement learning approaches require a massive amount of data collected in the environment to work, which slows down its success in tasks where data collection is time-consuming or costly, like robot manipulation. Model-based reinforcement learning (MBRL) methods alleviate this issue by maintaining a world model that simulates the real environment. The world model can serve as a surrogate of the real environment for the agent to interact with to reduce the amount of the required time-consuming interaction in the real environment. MBRL methods (Kaelbling et al., 1996; Wang et al., 2019; Janner et al., 2019) learn from model rollouts of previously observed states. Recently, CTRL (Lu et al., 2020) takes a structural causal model (SCM) approach that can generate samples counterfactually had a different action had been taken for a state previously observed. However, these methods are limited for robotic manipulation tasks since the environment is often the key limiting factor. In this paper, we perform counterfactual reasoning on the object properties. For example, when the task manipulates objects with different masses, the real environment may not have a uniform distribution of object masses. Furthermore, to avoid damaging the robot, certain exploration of the gripper torque may be limited during training.

To this end, we propose a Dyna-style MBRL method, CausalDyna in robotics that improves the policy performance by counterfactual reasoning of physics properties of objects and enriching the diversity of the generated rollouts. We leverage the structural causal model (SCM) to model the state dynamics. CausalDyna can be applied to generate episodes with unseen or rarely seen objects to improve the sample efficiency and generalization of the policy.

Our contributions are summarized as follows.

- We introduce a novel Dyna-style causal reinforcement learning algorithm, dubbed as CausalDyna that learns from counterfactually generated episodes with intervened object property values.
- We compare with state-of-the-art model based reinforcement learning algorithm, MBPO and model free algorithm, SAC on the CausalWorld environment. Experimental results

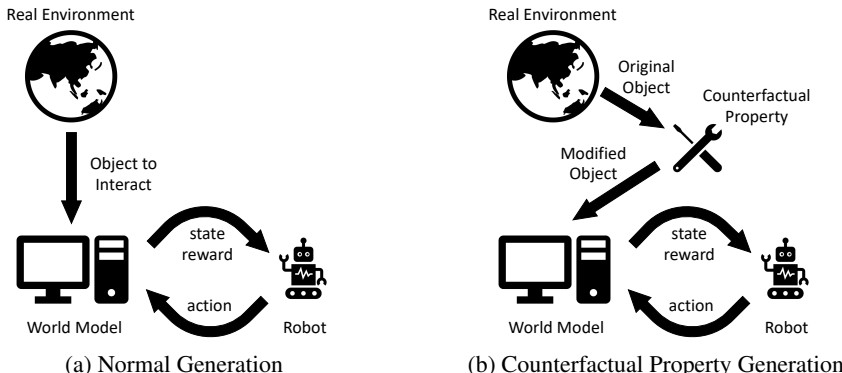

(a) Normal Generation        (b) Counterfactual Property Generation

Figure 1: In classical Dyna-style methods, the world model generates episodes starting from a real environment state. Then, our robot can practice in the world model and learn how to manipulate the original object. To improve the generalization of the learned policy, we further modified the object property in the state. So the robot has the chance to play with objects with more diverse properties.

show that CausalDyna outperforms MBPO and SAC on sample efficiency by up to 17% and generalization by up to 30% when manipulating objects with unseen or rarely seen properties.

## 2 RELATED WORK

**Causal Inference in Reinforcement Learning**   There is an increasing interest in causal inference in the field of reinforcement learning. Counterfactually-Guided Policy Search (CF-GPS) (Buesing et al., 2018) assumes that the real transition, observation, and reward functions are all known. They show that any partially observable Markov decision process (POMDP) can be represented as a structural causal model (SCM). Therefore, counterfactual inference can be applied to improve the off-policy evaluation and policy-guided search. CounTerfactual Reinforcement Learning (CTRL) (Lu et al., 2020) leverages bidirectional conditional GAN to model the environment dynamic for data augmentation. The model takes a noise vector as input besides the state and action to model the randomness of the environment. Before generating counterfactual data given alternative actions, they first infer the value of this noise vector. Then, the inferred noise is used to generate predictions with new actions. Causal Partial Models (CPM) (Rezende et al., 2020) studies the causal incorrectness of world models that don't condition on the full observation. To fix this issue, CPM introduces a backdoor variable that helps the rollout of the model to be causally correct. We propose an SCM framework to model the physics properties of objects across the temporal dimension. In addition, we show that generating episodes with counterfactual object properties helps improve the generalization of the learned policy.

**Model-Based Reinforcement Learning**   Model-based Reinforcement Learning (MBRL) approaches have shown a potential to improve the sample efficiency by a large margin compared to classical model-free approaches (Kaelbling et al., 1996; Wang et al., 2019). Autoencoder-based algorithms like World Models (Ha & Schmidhuber, 2018) and Dreamer (Hafner et al., 2019; 2020) use the world model to better represent the visual observation and faster the policy training. Policy Search with Backpropagation algorithms like PILCO (Deisenroth & Rasmussen, 2011; Deisenroth et al., 2013; Kamthe & Deisenroth, 2018) and GPS (Levine & Koltun, 2013; Levine & Abbeel, 2014; Montgomery & Levine, 2016) train the policy by maximizing the simulated return of the policy in the world model. Because the world model is differentiable, the policy can be directly trained by gradient descent. Shooting algorithms like PETS-RS (Chua et al., 2018) and MB-MF (Nagabandi et al., 2018) alleviate the receding horizon problem in model predictive control (MPC). Recent works include Ross & Bagnell (2012), MOPO, (Yu et al., 2020) and Morel (Kidambi et al., 2020) show that MBRL can work well in the offline RL setting. Unlike the traditional MBRL that approximates the local transition function, $L^3P$ (Zhang et al., 2021) builds the world model as a graph of states for better reasoning ability. Dyna-style algorithms (Sutton, 1990; 1991a;b) use the learned world model to roll out simulated episodes to reduce the demand for real data for policy training. As a

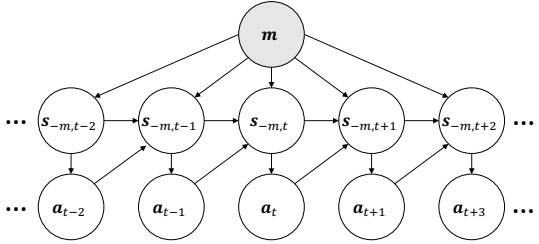

Figure 2: The structure causal model of a robot environment. The time-invariant property is modeled as a node $m$ across the temporal dimension that affects all the causal mechanisms. $s_{-m,t}$ and $a_t$ denotes the time-variant state and the action at the step $t$, respectively.

recent development of Dyna-style algorithms, ME-TRPO (Kurutach et al., 2018) uses an ensemble of world models to catch the epistemic uncertainty; MB-MPO (Clavera et al., 2018) viewed each model in the ensemble as a task and meta-learn a policy that adapts quickly to handle the model-bias issue; MBPO (Janner et al., 2019) rolls out short episodes branched from real data to improve the generation quality. Our method follows the Dyna-style framework and targets designing and using a causal world model to generate better and more diverse rollouts in robotic environments.

## 3 BACKGROUND

### 3.1 STRUCTURAL CAUSAL MODEL

Structural Causal Model (SCM) is a widely used framework to describe the causal mechanism of a system. Let's denote $\mathbb{X} = \{\mathbf{x}_1, ..., \mathbf{x}_N\}$ as the set of $N$ variables in a system. Knowing their causal relationships allows us to build a directed acyclic causal graph to describe this system. Each node represents a variable, which is directly caused by its parent nodes. In this way, a node $\mathbf{x}_n$ can be modeled as the following function:

$$\mathbf{x}_n = f_i(Pa_{\text{obs}}(\mathbf{x}_n), \mathbf{u}_n) \tag{1}$$

Here, $Pa_{\text{obs}}(\mathbf{x}_n)$ denotes the observed parent nodes of $\mathbf{x}_n$. $\mathbf{u}_n$ is a noise that represents the effect of omitted factors. This function is also called a causal mechanism. SCM is the set of these causal mechanisms that describes the whole system. SCM defines a joint distribution of the variables $p(\mathbf{x}_1, ..., \mathbf{x}_N)$ following the causal Markov assumption: given its direct causes, each variable $\mathbf{x}_n$ is independent of other indirect causal variables.

### 3.2 DYNA-STYLE MODEL-BASED REINFORCEMENT LEARNING

Dyna-style model-based reinforcement learning uses the world model to roll out simulated episodes, which can be viewed as data augmentation. The training of Dyna-style MBRL is composed of three steps: First, the agent interacts with the real environment and collects real data to train the world model. Then, this world model is used as a simulator of the real environment for the agent to interact and collect simulated data. After that, the agent can be trained together with the real and the simulated data using classical model-free reinforcement learning algorithms. These three steps are executed repeatedly until the training converges. In case we apply Dyna-Style algorithm on RL algorithms with experience-replay buffers and would like to collect whole simulated episodes, as the world model is trained to only approximate the transition of the environment $p(\mathbf{s}_{t+1}|\mathbf{s}_t, \mathbf{a}_t)$, we need an initial state to start the simulated episodes. A usual way to solve it is using the first state or a randomly sampled state $\mathbf{s}_t$ from the collected real episodes as the start point of the simulated episodes. As the real episode already contains the future of $\mathbf{s}_t$ under the original action sequence $\{\mathbf{a}_t, \mathbf{a}_{t+1}, ...\}$ executed in this episode, generating new simulated episodes starting from $\mathbf{s}_t$ under different action sequences can be viewed as answering a counterfactual "what if" question: What would happen if the agent behave differently this time instead of doing $\{\mathbf{a}_t, \mathbf{a}_{t+1}, ...\}$? The world model gives the agent a chance to figure out the answer without interacting in the real environment, and helps the agent learn faster.

---

**Algorithm 1:** Counterfactual Property Generation

---

**Data:** Rollout length $K$, Real experience buffer $\mathbb{D}_r$, Policy $p_\pi$, World model $p_{WM}$,
      Counterfactual property space $M$, Empty episode buffer $\mathbb{B}$

**Result:** $\mathbb{B}$

1   Sample a state $\boldsymbol{s} = [\boldsymbol{s}_{-m}; \boldsymbol{m}]$ from the real experience buffer $\mathbb{D}_r$, $\mathbb{B}$.append($\boldsymbol{s}$)

2   Sample a counterfactual property value $\boldsymbol{m}_{CF}$ from $M$, set $\tilde{\boldsymbol{s}} = [\boldsymbol{s}_{-m}; \boldsymbol{m}_{CF}]$

3   **for** $K$ *steps* **do**

4      $\tilde{\boldsymbol{a}} \sim p_\pi(\boldsymbol{a}|\tilde{\boldsymbol{s}})$, $\tilde{\boldsymbol{s}}' \sim p_{WM}(\boldsymbol{s}'|\tilde{\boldsymbol{s}}, \tilde{\boldsymbol{a}})$

5      $\mathbb{B}$.append($\tilde{\boldsymbol{a}}$, $\tilde{\boldsymbol{s}}'$), $\tilde{\boldsymbol{s}} \leftarrow \tilde{\boldsymbol{s}}'$

6   **end**

---

## 4   METHOD

### 4.1   STRUCTURE CAUSAL MODEL OF A ROBOT ENVIRONMENT

Let's consider an environment where a robot needs to manipulate an object. We can describe this environment using different states. Many of these states are changing over time, including the object position and the end-effect position. It is important to model them as they directly contain the dynamic information of the environment. Some other states are time-invariant, like the object mass or the floor friction coefficient. Although their values are fixed, they determine the environment dynamics and affect how other time-variant states change. Let's denote the total state at step $t$ as $\boldsymbol{s}_t$. $\boldsymbol{s}_t = [\boldsymbol{s}_{-m,t}; \boldsymbol{m}]$ is the concatenation of the time-variant state $\boldsymbol{s}_{-m,t}$ at step $t$ and the object time-invariant property $\boldsymbol{m}$. The motor torque to execute at step $t$ is denoted as $\boldsymbol{a}_t$. As shown in Fig.2, we can build a structural causal model (SCM) to describe this environment. The time-invariant property $\boldsymbol{m}$ is modeled as a fixed node across the temporal dimension, which affects all the causal mechanisms.

### 4.2   COUNTERFACTUAL PROPERTY GENERATION

Policy generalization ability is essential as the testing environment of the policy is not always the same as the training environment. For example, when learning to lift an object, the robot might only interact with objects whose masses are in a suitable range. Lifting frequently a too-heavy object might reduce its service life, and most reinforcement learning algorithms need a large amount of interaction data to work. However, knowing how to lift a heavy object is still desirable when deploying the robot. A typical Dyna-style method generates simulated rollouts branching from a starting state seen in previous real episodes. If the world model takes physics properties as input, it is possible to go a step further and intervene in these properties. For example, we could modify the mass of an object in the world model to make it heavier. So the agent can learn to manipulate them in the world model as much as we want without harming its service life. Inspired by this, we design a simple generation strategy to enrich the simulated rollouts by modifying the original object's property to improve the policy generalization. Concretely, instead of taking a starting state $\boldsymbol{s}_t = [\boldsymbol{s}_{-m,t}; \boldsymbol{m}]$ sampled from real episodes as it is like most of the Dyna-style methods, we replace the object property $\boldsymbol{m}$ by a desired counterfactual value $\boldsymbol{m}_{CF}$ sampled from a predefined counterfactual property space $M$ before rolling out the simulated episodes. We name this type of episodes generation as counterfactual property generation, illustrate it in Fig.1 and show the process in Alg.1.

### 4.3   TRAINING PROCEDURE

The training of our model follows the Dyna-style model-based reinforcement learning framework. The world model is an additional imperfect substitute for the real environment for the policy to interact with. The policy is still trained using the traditional model-free reinforcement learning approach, but the data for training is a mixture of the data from the real environment data and that from the world model. During the training procedure, we maintain two replay buffers. The real experience replay buffer $\mathbb{D}_r$ stores the interaction data from the real environment. The world model is trained using the real experience replay buffer only. The simulated episodes from the world model

---

**Algorithm 2:** Training Procedure

---

**Data:** Policy $p_\pi$, World model $p_{WM}$, Empty real experience replay buffer $\mathbb{D}_r$, Empty episode buffer $\mathbb{B}$, Rollout length $K$, Counterfactual property space $M$, Counterfactual generation ratio $\alpha$

**Result:** Trained Policy $p_\pi$

1 Prefill $\mathbb{D}_r$ by executing the untrained policy $p_\pi$ in the environment

2 **while** *Not Converge* **do**

3      Split $\mathbb{D}_r$ into a training set $\mathbb{D}_{r,train}$ and holdout set $\mathbb{D}_{r,holdout}$ randomly

4      Train the world model $p_{WM}$ on $\mathbb{D}_{r,train}$ until converge on $\mathbb{D}_{r,holdout}$

5      Empty the simulated experience buffer $\mathbb{D}_s$

6      Generate $\alpha\% \cdot N_f$ simulated episodes with $K$ steps by counterfactual property generation as Alg.1 to $\mathbb{D}_s$

7      Generate $(1 - \alpha\%) \cdot N_f$ simulated episodes with $K$ steps with original property to $\mathbb{D}_s$

8      **for** *E steps* **do**

9          Collect a step of data in the real environment; add it to $\mathbb{D}_r$

10          Update policy parameters via SAC on the combination of $\mathbb{D}_r$ and $\mathbb{D}_s$ for $G$ steps

11      **end**

12 **end**

---

are stored in the simulated experience buffer $\mathbb{D}_s$, which is used to train the policy net and the real experience replay buffer $\mathbb{D}_r$. The whole training procedure is shown in Alg.2. The policy is trained via soft actor-critic (SAC) (Haarnoja et al., 2018) using the data from both the real experience buffer $\mathbb{D}_r$ and the simulated buffer $\mathbb{D}_s$. As we generate the simulated episodes with counterfactual property and we following the Dyna-style MBRL framework, we name our model CausalDyna.

**World Model Training**    Each time the world model is trained, the real experience replay buffer $\mathbb{D}_r$ is split into a training set, and a holdout set randomly. The world model is trained to predict the next state $s_{t+1}$ by maximizing the log-likelihood given the current state $s_t$ and the action $a_t$ in the training set until converging measured by the holdout set.

**Augment Data Collection**    We adopt the generation strategy of model-based policy optimization (MBPO) (Janner et al., 2019) to roll out the world model. The simulated episodes start from a real state randomly sampled from the real experience replay buffer $\mathbb{D}_r$ and are rolled out for $K$ steps. We generate two types of simulated episodes: $\alpha\%$ of the rollouts are generated with counterfactual property generation, where we intervene the object property as described in Alg.1 to generate episodes with different objects. The remaining $(1 - \alpha\%)$ episodes are generated using the original property. Each time $N_f$ simulated episodes are generated in total. Note that each time we collect the simulated episodes, all the previous data in the simulated experience buffer $\mathbb{D}_s$ is discarded as the world model generated them a few training steps before and are not 'fresh' anymore.

## 5 EXPERIMENTS

### 5.1 BENCHMARK

We evaluate our method CausalDyna on a recently proposed robotic benchmark CausalWorld (Ahmed et al., 2020). CausalWorld is designed for causal structure and transfer learning in a robotic manipulation environment. The robot in CausalWorld is a 3-finger gripper. Each finger has three joints. The mission of the robot is to move objects to specified target locations. The observations of the CausalWorld we use includes the time stamp $t$, the robot state $s_r$, the object state $s_o$, the time-invariant property $m$, and the goal information $s_g$. The robot state $s_r$ is consists of 9 joint positions, 9 joint velocities, and the Cartesian coordinates of the three end-effectors (fingertips). The object state $s_o$ contains the Cartesian coordinate, the velocity, the quaternion orientation, and the object's angular velocity. The property $m$ includes the object mass and the friction coefficient. The goal information $s_g$ contains the target location and orientation of the object.

**Evaluated Models**  We evaluate three approaches in our experiments: Model-Based Policy Optimization (MBPO) (Janner et al., 2019), one of the state-of-the-art Dyna style methods with high sample efficiency, Soft Actor-Critic (SAC) (Haarnoja et al., 2018), a widely-used model-free approach, and our method CausalDyna.

**Task Settings and Performance Metrics**  We define three settings to evaluate our method: Picking Mass, Pushing Mass, and Pushing Friction. In Picking Mass, the robot needs to pick up an object to a target location in the air. The object mass is different over different episodes. In contrast, the target locations in Pushing Mass and Pushing Friction are on the ground. The object mass and the floor friction in Pushing Mass and Pushing Friction are different over different episodes, respectively. We use the default reward signals of CausalWorld to train our method. The reward provides rich signals to encourage the robot to get close to the object and move it toward the target. The reward is a weighted sum over the reduction of the distance between the end effectors and the object and the distance between the object and the target. We evaluated our approach and competing methods using fractional success rate (FSR), which is defined as the overlapping ratio between the object and the target. We compute the FSR of a given episode as the average FSR over the last 20 steps. To quantify the sample efficiency in our benchmark, we propose a metric named Area-Under-the-Curve Ratio (AUCRatio). Given a learning curve FSR $= f_{learn}(n_{step})$ where $n_{step}$ denotes the number of the environment steps collected already, AUCRatio until step $N_{step}$ is computed as Eq.2. As $0 \leq$ FSR $\leq 1$, a policy with AUCRatio $= 1$ means it can perform the task perfectly without training.

$$\text{AUCRatio} = \frac{1}{N_{step}} \sum_{n_{step}=1}^{N_{step}} f_{learn}(n_{step}) \tag{2}$$

## 5.2 Experiments with Out-of-Distribution Property

An intelligent robot might encounter various objects when deploying. If the robot need to manipulate an object unseen during training, its performance might be reduced. This can be viewed as an out-of-distribution problem: how to generalize well to the object not in the training distribution? The counterfactual property generation approach has the potential to increase the performance on objects with unseen property values if we roll out simulated episodes with object property that is out of the training range. To verify our assumption, we create an experiment to study whether our method helps improve the agent performance on objects whose property value is not encountered during training. In detail, in our Picking Mass and Pushing Mass setting, the robot is trained with objects of which the mass is uniformly distributed from 0.015kg to 0.045kg. But during the testing stage, the robot is asked to interact with heavier objects up to 0.1kg. In Pushing Friction setting, the friction coefficient is from 0.3 to 0.6 during training. And the robot is deployed to also handle friction from 0.6 to 0.8.

As we target the performance of the objects with unseen property value during training, we use our method here to imagine these objects. In detail, when the counterfactual property generation is applied, we replace the original property value with a counterfactual value uniformly sampled from the unseen test range. In this way, our agent can practice manipulating these unseen objects in the world model in advance.

**Hyperparameters**  The length of the simulated episodes $K$ is 10. A bootstrap ensemble of world models is used following Kurutach et al. (2018) for both MBPO and our method. The ensemble size is 7. For each generation step, we randomly pick one model from the ensemble to predict the next state. When training the policy, 20% of the training data are from the real experience replay buffer. The remaining are from the simulated episodes. In our CausalDyna, 20% of the simulated episodes are generated by counterfactual property generation ($\alpha$ in Alg.2). We use Adam (Kingma & Ba, 2014) as the training optimizer for all experiments. All the models we evaluated are trained for 1.2 million steps in Picking Mass and 600 thousand steps in Pushing Mass and Pushing Friction. Each model in this experiment has 5 training cases. The model architecture and the remaining hyperparameters can be found in Appx.A and Appx.B.

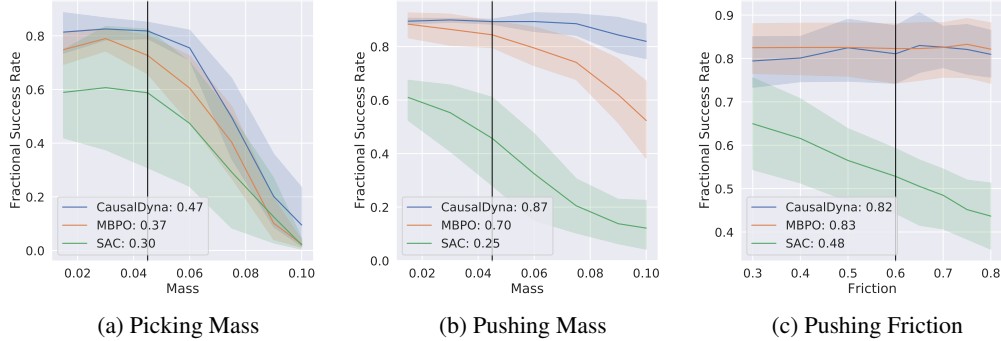

(a) Picking Mass      (b) Pushing Mass      (c) Pushing Friction

Figure 3: Experimental results of counterfactual property generation in the out-of-distribution experiment. The vertical black line shows the boundary between the seen and unseen property values during training. The left part is the seen region. Counterfactually generating the simulated episodes with unseen property value helps alleviate the performance drop when evaluating unseen property during training. Numbers in the legend denote the average performance in the unseen value range. Each curve contains 5 training cases.

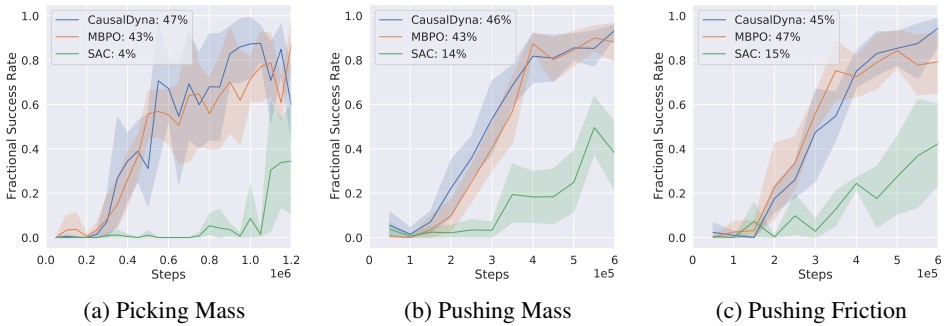

(a) Picking Mass      (b) Pushing Mass      (c) Pushing Friction

Figure 4: The learning curve of the evaluated models on the training property range in the out-of-distribution experiments. CausalDyna converges as fast as MBPO, although it generates 20% less simulated episodes in the training property range. Numbers in the legend denote the average AUCRatio.

**Performance** The experimental results are shown in Fig.3. The vertical black line denotes the boundary between the seen and unseen values during training. The left part is the seen region. In Picking Mass and Pushing Mass, the performance of all the methods declines when the object mass is out of the training range. Moreover, the performance reduction is more significant when the tested object mass is farther away from the training range. Our CausalDyna alleviates this performance reduction in the unseen range by a large margin compared to MBPO. In Picking Mass, CausalDyna improves the unseen FSR by 24% from 0.37 to 0.47. For Picking Mass it is 27% from 0.7 to 0.87. This indicates that hallucinating episodes with unseen objects during training helps improve the generalization ability of the policy. In Pushing Friction, CausalDyna achieves similar performance as MBPO since the unseen range performance reduction here is not obvious. As SAC is less sample efficiency than both model-based methods, SAC cannot achieve compatible results given the same training data as MBPO and CausalDyna. Note that in Picking Mass, although our CausalDyna performs better than MBPO in the out-of-distribution range, the absolute performance is not high when the object is too heavy (like 0.1kg). This might be caused by the reduced performance of the world model when counterfactually generating episodes with unseen objects. A better world model design that can better understand the physics and reason the future more causally might help alleviate this issue when combined with our method. We leave this for future research.

**Sample Efficiency** We show the learning curve of MBPO, CausalDyna, and SAC of this experiment in Fig.4. Although we augment 20% fewer simulated episodes in the original property range

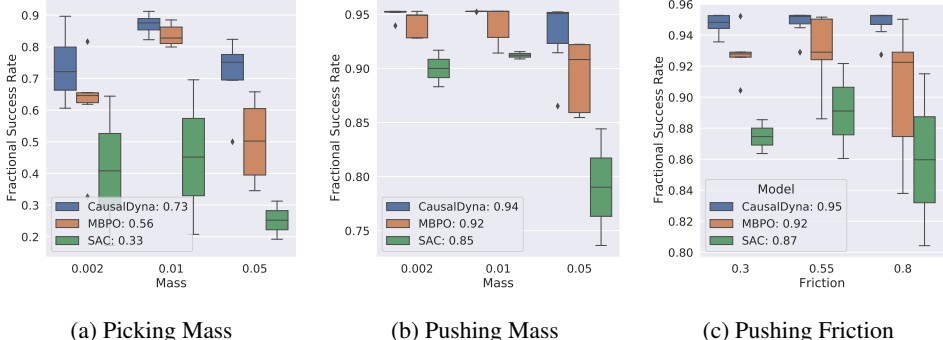

(a) Picking Mass          (b) Pushing Mass          (c) Pushing Friction

Figure 5: Experimental results of counterfactual property generation in the unbalanced distribution experiment. When counterfactually generating episodes where the object is less encountered during training, CausalDyna helps improve the policy performance on both the objects with head values and tail values. For each property, the median value occurs 90% of the time in the environment, and the rest two values share the remaining 10% equally. Numbers in the legend denote the average performance over the tail values. Each model has 6 training cases.

compared to MBPO, CausalDyna converges as fast as MBPO in the original training range. Results indicate that our method improves the out-of-distribution performance without sacrificing the sample efficiency. The model-free SAC training is much slower than MBPO and CausalDyna, as SAC doesn't have simulated data to train on.

## 5.3 EXPERIMENTS WITH UNBALANCED TRAINING DISTRIBUTION

In real environments like warehouses, the numbers of different wares are unequal, and a sorting robot might manipulate some objects less frequently. This can be described as an unbalanced training distribution. If the training distribution is heavily unbalanced and some objects are significantly less encountered than others during training, counterfactually generating episodes with such objects might help improve the policy performance on them. We create a simple heavily unbalanced training distribution consisting of 1 head property value and two tail property values to verify this assumption. The object property in 90% of the training episodes equals the head value. The two tail values share the remaining 10%, each value obtains 5%. Concretely, in Picking Mass and Pushing Mass, we have three different objects with mass values 0.002kg, 0.01kg, and 0.05kg, respectively. 90% of the time, the robot sees and manipulates the object with the median mass value of 0.01kg. The robot plays with the heavy 0.05kg object and the light 0.002kg object equally in the remaining time. For Pushing Friction, the three friction coefficients are 0.3, 0.55, and 0.8 that occur in 5%, 90%, and 5% of the time, respectively. In the testing stage, models need to perform well on all three property values.

As the objects with tail values occur less frequently in the training stage, CausalDyna in this experiment imagines what would happen if the given head object is the tail. Concretely, when CausalDyna generating simulated episodes, the property value of original objects are counterfactually modified to one of the tail property values randomly. Therefore, the agent can interact with the tail objects more in the world model to improve the tail performance.

**Hyperparameter** In CausalDyna, 2/3 of the simulated episodes are generated by counterfactual property generation ($\alpha$ in Alg.2). All the models on all the 3 settings are trained for 600 thousand steps. Each model in this experiment has 6 training cases. The remaining hyperparameters are the same as in the previous experiment.

**Performance** As shown in Fig.5, the performance on the head property value (0.01kg for mass and 0.55 for friction) is better than the tail property values for all the methods in all the 3 settings. However, CausalDyna improves the performance on the tail property and shows the smallest performance difference between the head and the tail among the three models. For example, the performance gap between the head and the tail of CausalDyna in Picking Mass is about 0.1, much smaller than MBPO

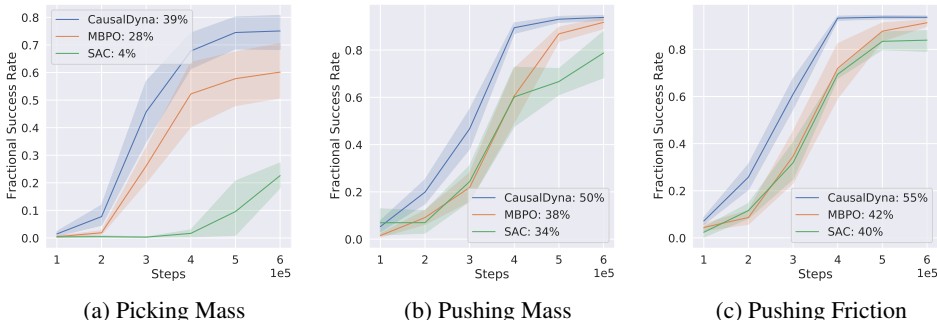

| (a) Picking Mass | (b) Pushing Mass | (c) Pushing Friction |

Figure 6: Average policy performance at different environment steps. Our method CausalDyna, which counterfactually generating episodes where the object is less frequently encountered during training, reduces the required amount of environment steps and shows the best sample efficiency in the unbalanced training distribution experiment. Numbers in the legend denote the average AUCRatio. Each model has 6 training cases.

(0.2-0.3), and the tail performance is increased by 30% from 0.56 to 0.73. Besides, we notice that CausalDyna improves the policy performance on both objects that are less frequently seen during training and the head objects compared to MBPO. This might be because learning how to behave well in the tail cases helps the model better understand the environment dynamics and improves overall performance. In addition, the performance variance in Pushing Mass and Pushing Friction of CausalDyna is much lower than the other two methods, which suggests that the performance of CausalDyna is more consistent than other methods. With the same amount of training data as MBPO and CausalDyna, the model-free SAC's performance is worse than the model-based MBPO and CausalDyna, which is the same as the out-of-distribution experiment.

**Sample Efficiency**    The learning curves of the evaluated models are shown in Fig.6. The fractional success rate is uniformly averaged over all the property values. CausalDyna shows a better sample efficiency and converges faster. In all three settings, CausalDyna requires about 100k fewer environment steps to converge compared to MBPO and increase the sample effiency by about 17%. This might be because CausalDyna has more simulated episodes with the tail property values to train the agent, which helps the agent understand the task better and adapt to all the property values faster.

## 6    CONCLUSION AND FUTURE WORK

In this paper, we focus on improving the generalization ability of model-based reinforcement learning in robotic environments. We propose a novel Dyna-style causal reinforcement learning algorithm named CausalDyna that rollouts episodes with intervened object properties. CausalDyna leverages the diversity of the simulated episodes augmented by the world model and improves the generalization of the policy when manipulating objects with property unseen or rarely seen during training. Experiments show that our method helps the robot generalize to objects with unseen property values better. In addition, when the training distribution is unbalanced, our method requires fewer environment steps to converge and performs better with rarely seen objects.

To our knowledge, we are the first to propose counterfactual reasoning on environment properties to improve the generalization of reinforcement learning. We believe this is a promising direction to solve many complex reinforcement learning tasks where the policy generalization ability is essential. When combined with model predictive control and counterfactual reasoning on actions, it is possible to further improve sample efficiency and generalization of RL algorithms. One limitation of our method is that the quality of our counterfactual episodes depends on how well our world model understands the environment. We plan to design a better world model that takes prior knowledge like simple physics laws into account. Finally, we have assumed that the properties in our environment are fully observable in our current work. We plan to investigate causal models with latent variables representing unobserved properties of the environment.

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

# A    MODEL ARCHITECTURE

Here we list the architecture of the world model, the policy actor net and the policy critic net we use in all experiments for all methods. All the models are built using linear-layers. The world model uses Swish activation function (Ramachandran et al., 2017) and the policy uses ReLU (Nair & Hinton, 2010).

Table 1: Model Architecture

| Modules | Hidden Layers | Neurons Per Layer |
|---|---|---|
| World Model | 3 | 200 |
| Policy Actor | 2 | 256 |
| Policy Critic | 2 | 256 |

# B    HYPERPARAMETER

The size of the real experience replay buffer $\mathbb{D}_r$ is 100k for MBPO and our method CausalDyna in all three settings. For SAC, it is 1M as we notice SAC with 100k-size replay buffer cannot be trained well. For the world model training, The replay buffer $\mathbb{D}_r$ is split randomly into a training set $\mathbb{D}_{r,train}$ with 80% of the data and a holdout set $\mathbb{D}_{r,holdout}$ containing the remaining data. We train the model once for every 250 real environment steps until converge is evaluated on the holdout set. The learning rate is 3e-4. Batch size is 256. For the policy training, the policy net is updated for 5 iterations per real environment step. The batch size is 256, and the learning rate is set to 1e-4.

# C    QUALITATIVE RESULTS

Here we demonstrate episodes from CausalDyna and MBPO in the Pushing Mass setting in unbalanced training distribution experiments with the heavy tail object in Fig.7 and Fig.8. Both models are trained for 600k environment steps. The object to manipulate is in blue color. Target location is shown as the green shade. Each column corresponds to an episode. CausalDyna generalizes to the heavy tail object well and pick it to the location successfully shown in Fig.7, while MBPO fails to lift the object up in 2 episodes shown in Fig.8.

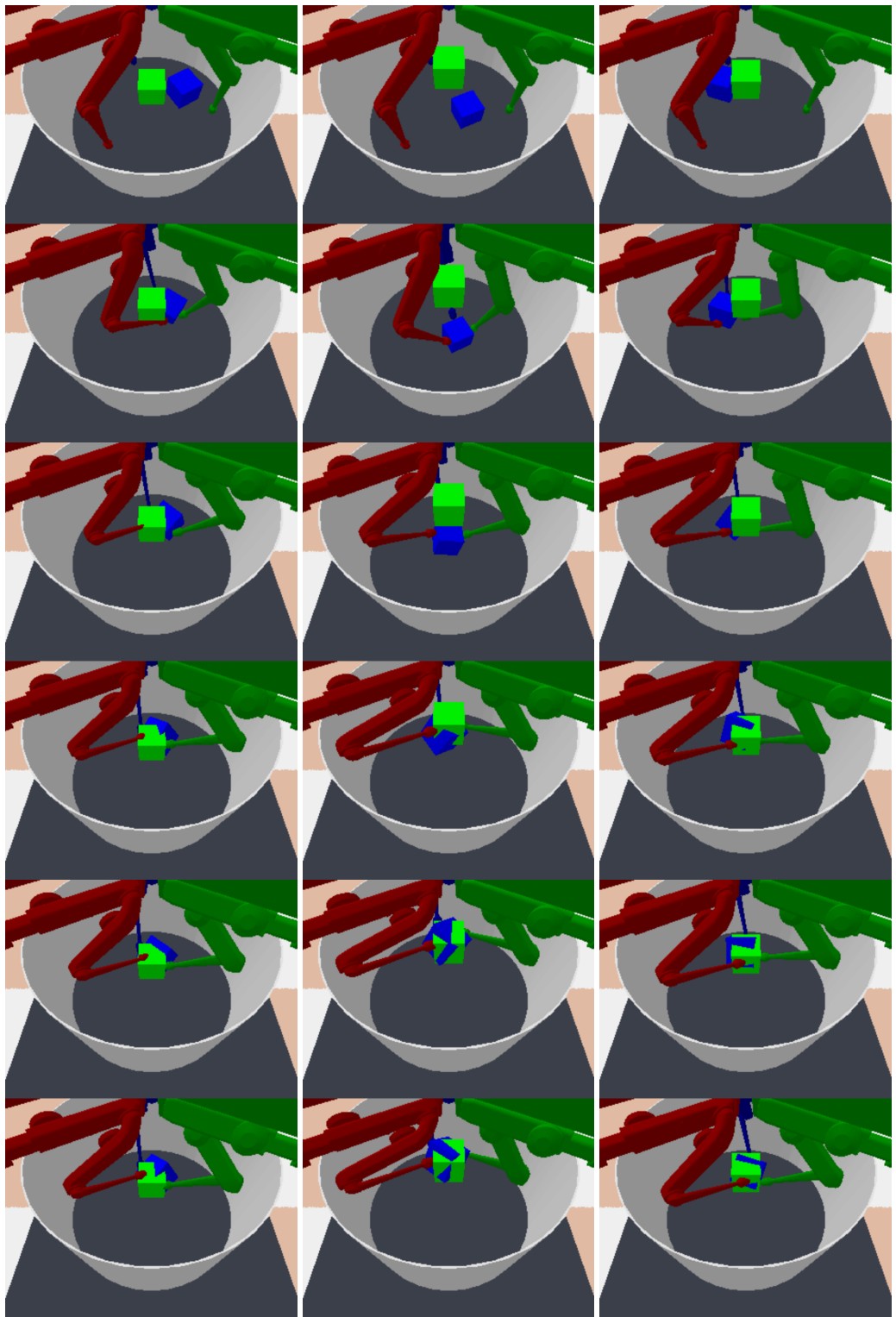

Figure 7: CausalDyna with the heavy tail object. Pushing Mass, Unbalanced Training Distribution.

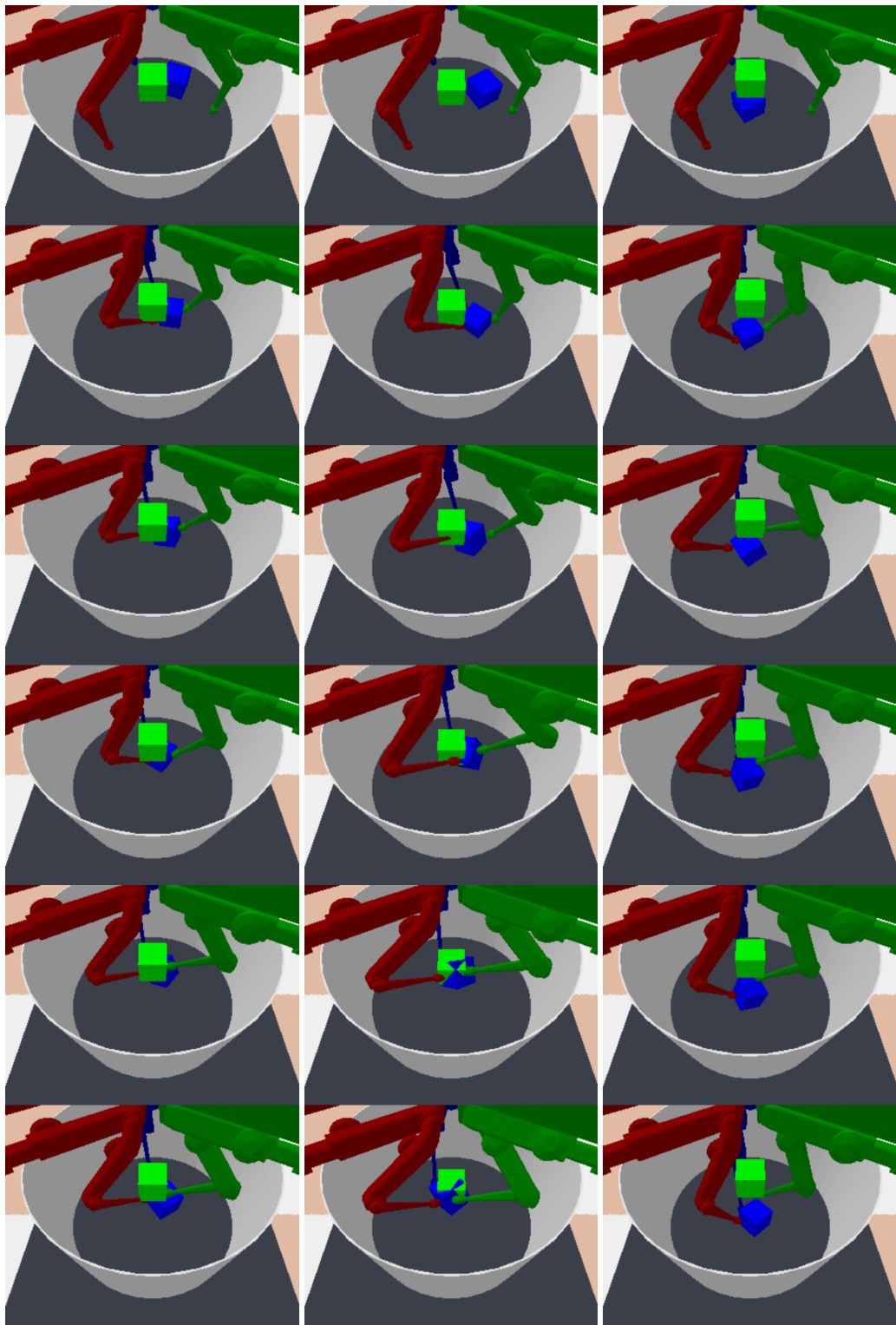

Figure 8: MBPO with the heavy tail object. Pushing Mass, Unbalanced Training Distribution.

