# OpenReview forum: "CausalDyna: Improving Generalization of Dyna-style Reinforcement Learning via Counterfactual-Based Data Augmentation"
_ICLR.cc/2022/Conference — ICLR 2022 Submitted_

### Official Review · Reviewer_VSvt · 2021-10-27

**Correctness:** 2
**Technical Novelty And Significance:** 2
**Empirical Novelty And Significance:** 1
**Recommendation:** 3
**Confidence:** 5

**Main Review:**

Enhancing the diversity of training data (here, time-invariant properties of the state) to improve the robustness and extrapolation capacities of an RL agent is a good idea but is standard practice known as domain randomization.  In the second application, countering data imbalance, this amounts to conventional data up-/downsampling.

This paper motivates data augmentation / domain randomization and data up-/downsampling in terms of causality theory.  However, this perspective provides no additional insight; the structural causal model it introduces does not appear to be used by the method at all.  Moreover, the presentation does not cleanly separate counterfactual reasoning from intervention.  Altering state is a form of intervention; calling it counterfactual because such state was not seen in real data dilutes the notion of counterfactual reasoning (is all simulated training counterfactual?).

Methodologically, the greatest weakness of the method, acknowledged by the authors, is that there is no way to train the model on altered data.  Thus, the performance of the policy on these altered data hinges on the extent to which the model, trained without such data, happens to make accurate predictions.  In other words, if the trained model does not extrapolate, the entire method is ineffective.

The method is compared to two representative baselines, but the most important baseline is absent: an ablated CausalDyna that does not alter any state.

Some Details:

- Figure 1 is not referred to in the text.

- Section 3.2 misrepresents Dyna by separating model training on real data, collecting simulated experience, and training the value function / policy into three distinct steps.  In the original formulation, there is no experience-collection phase, and model training and value function / policy training on real and simulated data proceed in a parallel or interleaved fashion.  The separate data-collection phase only makes sense with RL algorithms that use experience-replay buffers.

- Why is the FSR averaged over the last 20 steps? Are the trained policies unable (or not trained) to bring the object to rest?

- In Eqn. 2 the factors of 1 in the numerator and denominator are superfluous.

- "5 training cases": Only later it becomes implicitly clear that this is intended to mean 5 independent runs from which means and (presumably) standard deviations are computed.

**Summary Of The Paper:**

This paper addresses the problem of out-of-distribution states encountered in reinforcement learning, and proposes a data augmentation method for Dyna-style algorithms that amounts to domain randomization. The central idea, altering elements of the state vector while training on the model, is motivated from a causality perspective.


**Summary Of The Review:**

- The paper claims to use structural causal models but this is the case only in a trivial sense at best.

- The causality perspective provides no novel insight.

- The experimental setup pales in comparison to other uses of domain randomization.

---

> ### Author Response · Authors · 2021-11-19
> **Reply**
>
> Thank you for your detailed and helpful review, questions and suggestions. We here address them.
>
> **Enhancing the diversity of training data to improve the robustness and extrapolation capacities is a standard practice known as domain randomization.**
>
> Our method is different from domain randomization. Normal domain randomization assumes an environment where we can directly modify the state to randomize the setting, like a carefully designed dexterous hand simulator based on some physics engines like Bullet or MuJoCo. In our case, we don't have such a changeable environment for us to randomize the domain. Instead, we need to train a black box world model to mimic the original environment from scratch intervene in the state in our learned world model. Our setting is harder than the normal domain randomization.
>
> **Countering data imbalance, this amounts to conventional data up-/downsampling.**
>
> Our method is not up-/down sampling. Instead of using the sampling strategy to balance the data, we still keep the original data distribution and intervene in the head data state to the tail to balance the data.
>
> **The most important baseline is absent: an ablated CausalDyna that does not alter any state.**
>
> This baseline is the MBPO model in our experiments.
>
>
> **The structural causal model it introduces does not appear to be used by the method at all. Does not cleanly separate counterfactual reasoning from intervention**
>
> We use SCM to model and analysis the task we target to solve. We plan to make the properties unobservable and use SCM to infer both the environment randomness and the unobserved properties as the abduction in counterfactual reasoning in the next step.
>
>
> **If the trained model does not extrapolate, the entire method is ineffective.**
>
> As the property goes far away from the training range, the extrapolation ability of the world model goes down and leads to the decrease of the policy performance of our method as shown in Fig.3 (a,b). However, Fig.3 (a,b) also shows that our method can utilize the extrapolation ability of the current world model and alleviate the performance drops compared to MBPO. We agree that designing a model that can better extrapolate is important and we will focus on it as the next step.
>
>
> **Section 3.2 misrepresents Dyna. There is no experience-collection phase, and model training and value function / policy training on real and simulated data proceed in a parallel or interleaved fashion. The separate data-collection phase only makes sense with RL algorithms that use experience-replay buffers.**
>
> Thanks for pointing this out! Dyna algorithm needs to collect simulated data (maybe just one step transition in the simulator) and use the simulated and real data to update the policy/value function in parallel or interleaved fashion. We think the simulated experience collection stage is still here. We agree that our original statement is not accurate, as we said that the method collects 'a large number of simulated episodes' in this stage, which is indeed only suitable for the RL with experience-replay buffers. We revised this section to make our statement more accurate with the help of your suggestions.
>
> **Why is the FSR averaged over the last 20 steps? Are the trained policies unable (or not trained) to bring the object to rest?**
>
> FSR averaged over the last 20 steps is one of the default metrics from the benchmark we use. As the task is to move the object to a target position, this metric checker whether the robot completes the task at the end of the episode. So, the robot in a given episode cannot successfully move and keep the object in the desired target position, this episode is judged as a failure.  To smooth the result, this metric average over the last 20 steps instead of only using the last time step.
>
> **In Eqn. 2 the factors of 1 in the numerator and denominator are superfluous.**
>
> Thanks! The original factors of 1 in the equation are used for a quick understanding. We update Eq.2 to incorporate this suggestion
>
> **"5 training cases": Only later it becomes implicitly clear that this is intended to mean 5 independent runs from which means and (presumably) standard deviations are computed.**
>
> 5 training cases' means that we launch the training 5 times independently to obtain 5 trained models. Then, we evaluate these 5 models separately and average their score to compute the means and the 95\% confidence interval
>
> **Figure 1 is not referred to in the text.**
>
> We updated the paper and refer to our teaser figure 1 in Sec.4.2 now

---

> > ### Comment · Reviewer_VSvt · 2021-11-20
> > **Reply to Reply**
> >
> > Thank you for your detailed answer. Most importantly, your first comment (and Reviewer CWLF) convinced me that your take on data augmentation is indeed interesting. Also, thanks for clarifying that MBPO does in fact constitute an ablated CausalDyna. I suspected this but did not find confirmation when reading the paper.

---

### Official Review · Reviewer_cJzu · 2021-11-01

**Correctness:** 3
**Technical Novelty And Significance:** 2
**Empirical Novelty And Significance:** 3
**Recommendation:** 5
**Confidence:** 4

**Main Review:**

Strengths
- The idea of the paper is neat and intuitive. Improving generalization ability via causal learning has been a growing research area. This paper provides a solid demonstration of its promise in the context of model-based RL.
- The paper is very well written. The experimental analysis is rich and sound.

Weaknesses
- The claimed connection between the SCM and the proposed dynamic model seems vague. From what I've understood, the proposed method is essentially pretty much the same as the standard dynamic model with state inputs, and then performs interventions over a particular variable. It is not clear how SCM comes into play.
- The assumption made in the paper is very strong. It's known that learning an SCM (or a dynamic model that allows for counterfactual intervention) is a very challenging task by itself. Yet, the paper assumes it can be easily solved. In particular, it assumes
    - full access to the high-level variables (e.g., `object mass` and `friction coefficient`) in the true generative process
    - the learned dynamic model can generalize to unseen settings within a `predefined counterfactual property space`\
  These assumptions do not look realistic in the real-world.

**Summary Of The Paper:**

The paper presents a method to improve the generalization of model-based RL by means of interventional data augmentation. The key idea is to intervene the value of a particular variable (e.g., object property) in the learned dynamic model for episode simulations. Experimental results show that it improves (i) the generalization ablity in the OoD scenarios with respect to the intervened variable, (ii) sample efficiency in the presence of unbalanced training distribution.

**Summary Of The Review:**

The paper well demonstrates the benefits of counterfactual data augmentation for model-based RL. However, the technical contribution seems limited and involves very strong assumptions. I, therefore, consider it below the acceptance threshold.

---

> ### Author Response · Authors · 2021-11-19
> **Reply**
>
> We thank Reviewer cJzu for the valuable and helpful comments. We incorporated the feedback here.
>
> **It is not clear how SCM comes into play. It assumes full access to the high-level variables**
>
> We use SCM to model and analysis the task we target to solve. We plan to make the properties unobservable and use SCM to infer both the environment randomness and the unobserved properties as the abduction in counterfactual reasoning in the next step.
>
> **The learned dynamic model can generalize to unseen settings within a predefined counterfactual property space. These assumptions do not look realistic in the real-world.**
>
> Indeed, the performance of our method depends on the extrapolation ability of the world model. As the property goes far away from the training range, the extrapolation ability of the world model goes down and leads to the decrease of our policy performance as shown in Fig.3 (a,b). However, Fig.3 (a,b) also shows that our method still alleviates the performance drops compared to MBPO and as the neural-network-based world model still have some (limit) extrapolation ability and our method can benefit from it. We agree that designing a model that can better extrapolate is important and we will focus on it as the next step.

---

### Official Review · Reviewer_CWLF · 2021-11-02

**Correctness:** 2
**Technical Novelty And Significance:** 2
**Empirical Novelty And Significance:** 2
**Recommendation:** 3
**Confidence:** 4

**Main Review:**

Strengths:
- connects data augmentation to counterfactual property generation
- clearly written
- What is novel about applying counterfactual data augmentation to DYNA, as opposed to standard data augmentation techniques in other areas of machine learning (e.g. computer vision) is that the goal of counterfactual data augmentation should be to learn a model that is _equivariant_ to data augmentation, whereas standard data augmentation techniques mostly focus on making the model _invariant_ to data augmentation. Therefore, this is a novel problem setting worth studying.

Weaknesses
- The authors predefine the counterfactual property space $M$, which avoids the hard problem of learning the counterfactual property space from experience. But this is a very restrictive assumption: it's not merely that the authors assume that the mass properties are observable, the authors assume the entire range of masses is observable as well. Because the authors assume access to this counterfactual property space, their empirical experiments amount to comparing a method with additional data augmentation to methods that do not have this privileged data augmentation, which seems to not be an interesting question to ask, since it is well known that data augmentation, especially if you have knowledge of what the augmentations should be, would help performance.
-  What would be a more interesting question to ask is, given that we know data augmentation would help, how can we supervise the world model to be equivariant to different data augmentations? However, the paper does not discuss how they supervise the world model to make the correct prediction for different values of $m_{cf} \in M$. In particular, if $m_{cf}$ is intended to take on values of out-of-distribution $m$ from the perspective of the policy, how can we guarantee that the model will actually learn to be equivariant to the data augmentation to generate useful data for training the policy? This seems to be the most crucial point, because the paper seems to assume that the world model will make the correct prediction, but it is not discussed how the paper supervises the world model to do so.
- Figure 5 does seem to suggest that the world model does seem to predict slightly better with the proposed data augmentation, but why would it do so without supervision on the correct prediction for different values of $m_{cf} \in M$? This would be an interesting question to investigate, but this investigation and analysis is missing from the paper.
- On the other hand, Figure 3 and 4 seem to suggest that the proposed data augmentation does not help at all compared to MBPO (the error bars overlap enormously). This result seems to refute the author's hypothesis that counterfactual data augmentation is crucial, especially since in Figure 6 MBPO does not perform that much worse than CausalDyna.


**Summary Of The Paper:**

This paper considers the problem of generalization of a dynamics model to different versions of the same environment, indexed by parameter m. For example, this parameter $m$ might be the mass of an object to be picked up. The problem that the authors tackle is to generalize the agent's policy to environments with either different parameters $m$ or a different distribution of parameters $m$. The solution the authors propose is to randomly sample different values of $m$ during training of the model, as opposed to using the $m$'s observed from the agent's rollouts. Empirically we observe that the proposed method performs similar to MBPO in generalization to out-of-distribution $m$ (Section 5.2) and that generalizes better on heavier masses than MBPO in the unbalanced distribution setting (Section 5.3).

**Summary Of The Review:**

The problem setting is good, but the main conceptual weakness is a lack of explanation for how the model is supervised to be equivariant to different data augmentations. Furthermore, empirical results seem to suggest that the proposed data augmentation does not have much of an effect on performance, which seems to undermine the motivation for this paper.

EDIT (AFTER REBUTTAL)
Thank you to the authors for your response. Based on the authors' response, it appears that the majority concerns I had raised will be addressed in the next iteration of the paper. Therefore, I maintain my original score and look forward to the new and improved version.

---

> ### Author Response · Authors · 2021-11-19
> **Reply**
>
> Thank you for your helpful review, questions, and suggestions.
>
> **It's not merely that the authors assume that the mass properties are observable, the authors assume the entire range of masses is observable as well.**
>
> Our current project target a setting that when we have some objects with known properties, how we can generalize better to objects of which the properties are unseen during training.
> We plan to make the properties unobservable and use SCM to infer both the environment randomness and the unobserved properties as the abduction in counterfactual reasoning in the next step.
>
>
> **How can we supervise the world model to be equivariant to different data augmentations?  the paper seems to assume that the world model will make the correct prediction**
>
> Indeed, the performance of our method depends on the extrapolation ability of the world model. As the property goes far away from the training range, the extrapolation ability of the world model goes down and leads to the decrease of our policy performance as shown in Fig.3 (a,b). However, Fig.3 (a,b) also shows that our method still alleviates the performance drops compared to MBPO and as the neural-network-based world model still have some (limit) extrapolation ability and our method can benefit from it. We agree that designing a model that can better extrapolate is important and we will focus on it as the next step.
>
>
> **Figure 5 does seem to suggest that the world model does seem to predict slightly better with the proposed data augmentation, but why would it do so without supervision on the correct prediction for different values of $m_{cf} \in M$?**
>
> As the relationship between these physics properties and the environment is mostly monotonous (like the larger the object mass is, the harder the robot need to push), it is no surprise that the simple neural-network-based world model will have a limited extrapolation ability to the out-of-range values once it catches the simple monotonous relationship in the training data.
>
>
> **Figure 3 and 4 seem to suggest that the proposed data augmentation does not help at all compared to MBPO (the error bars overlap enormously), refute the author's hypothesis that counterfactual data augmentation is crucial, especially since in Figure 6 MBPO does not perform that much worse than CausalDyna.**
>
> Fig.3(a) shows that our method performs better than MBPO in the mass range from 0.04-0.06 (as the error bars almost don't overlap here). When we further increase the mass, the advantage of our method decreases due to the extrapolation ability of the world model. Fig.3(b) shows that our method is better in all the out-of-range values. Although MBPO in Fig 6 does not perform that much worse, we are clearly more sample efficient in Fig 6. We think improving the extrapolation ability would help our performance and will focus on it as the next step.

---

> > ### Comment · Reviewer_CWLF · 2021-11-30
> > **Response to rebuttal**
> >
> > Thank you to the authors for your response. Based on the authors' response, it appears that the majority concerns I had raised will be addressed in the next iteration of the paper. Therefore, I maintain my original score and look forward to the new and improved version.

---

### Official Review · Reviewer_RJye · 2021-11-03

**Correctness:** 3
**Technical Novelty And Significance:** 2
**Empirical Novelty And Significance:** 2
**Recommendation:** 5
**Confidence:** 4

**Main Review:**

In general, the paper is well presented, and I can easily grasp the main idea. The major concern lies in the novelty, I think building an environment with a counterfactual technique is not new, which has been proposed before. This paper did not provide enough increments on the previous methods. In addition, I would like to ask how to handle the error produced by the SCM, maybe theoretical analysis can be provided to demonstrate the bound of the agent when being trained on such error-involved SCM.

**Summary Of The Paper:**

This paper aims to design a counterfactual reinforcement learning model to improve the training efficiency of the agent. To achieve this goal, SCM is leveraged to simulate the environment and generate new counterfactual trajectories. To demonstrate the effectiveness of the proposed model, the authors have conducted many experiments.

**Summary Of The Review:**

Not quite ready for top conferences like ICLR

---

> ### Author Response · Authors · 2021-11-19
> **Reply**
>
> Thank you for your helpful review, questions, and suggestions.
>
> **Building an environment with a counterfactual technique is not new, which has been proposed before. Did not provide enough increments on the previous methods**
>
> Thank you for this comment. We will focus on building a model that can better extrapolate, making the properties unobservable, and modeling the environment randomness into SCM as our next step.
>
> **How to handle the error produced by the SCM.**
>
> Indeed, the performance of our method depends on the extrapolation ability of the SCM. As the property goes far away from the training range, the extrapolation ability goes down and leads to the decrease of our policy performance as shown in Fig.3 (a,b). However, Fig.3 (a,b) also shows that our method still alleviates the performance drops compared to MBPO and as the neural-network-based world model still have some (limit) extrapolation ability and our method can benefit from it. We agree that designing a model that can better extrapolate is important and we will focus on it as the next step.

---

### Decision · Program_Chairs · 2022-01-20

**Decision:**

Reject

**Comment:**

The paper describes a new method to improve the generalization of model-based RL by means of interventional data augmentation. The key idea is to intervene the value of a particular variable (e.g., object property) in the learned dynamic model for episode simulations. Experimental results show that it improves (i) the generalization ablity in the OoD scenarios with respect to the intervened variable, (ii) sample efficiency in the presence of unbalanced training distribution.

Strengths:

- connects data augmentation to counterfactual property generation
- clearly written
- novel about applying counterfactual data augmentation to DYNA, as opposed to standard data augmentation techniques in other areas of machine learning
- The paper well demonstrates the benefits of counterfactual data augmentation for model-based RL

Weaknesses:

- a lack of explanation for how the model is supervised to be equivariant to different data augmentations.
- empirical results seem to suggest that the proposed data augmentation does not have much of an effect on performance
- The claimed connection between the SCM and the proposed dynamic model seems vague
- The technical contribution seems limited and involves very strong assumptions.
- The structural causal model it introduces does not appear to be used by the method at all.
- the presentation does not cleanly separate counterfactual reasoning from intervention
- he greatest weakness of the method, acknowledged by the authors, is that there is no way to train the model on altered data. Thus, the performance of the policy on these altered data hinges on the extent to which the model, trained without such data, happens to make accurate predictions

All the reviewers voted for rejection. I recommend the authors to use the reviewrs' comments to improve the paper and resubmit to another venue.